# The influence of HLA genotype on the development of metal hypersensitivity following joint replacement

David J. Langton [1✉], Rohan M. Bhalekar[1], Thomas J. Joyce[2], Stephen P. Rushton[2], Benjamin J. Wainwright[3], Matthew E. Nargol [1], Nish Shyam [1], Benedicte A. Lie[4], Moreica B. Pabbruwe [5], Alan J. Stewart [6], Susan Waller[7], Shonali Natu[7], Renee Ren[8], Rachelle Hornick[8], Rebecca Darlay[2], Edwin P. Su[8] & Antoni V. F. Nargol[7]

## Abstract

**Background** Over five million joint replacements are performed across the world each year. Cobalt chrome (CoCr) components are used in most of these procedures. Some patients develop delayed-type hypersensitivity (DTH) responses to CoCr implants, resulting in tissue damage and revision surgery. DTH is unpredictable and genetic links have yet to be definitively established.

**Methods** At a single site, we carried out an initial investigation to identify HLA alleles associated with development of DTH following metal-on-metal hip arthroplasty. We then recruited patients from other centres to train and validate an algorithm incorporating patient age, gender, HLA genotype, and blood metal concentrations to predict the development of DTH. Accuracy of the modelling was assessed using performance metrics including time-dependent receiver operator curves.

**Results** Using next-generation sequencing, here we determine the HLA genotypes of 606 patients. 176 of these patients had experienced failure of their prostheses; the remaining 430 remain asymptomatic at a mean follow up of twelve years. We demonstrate that the development of DTH is associated with patient age, gender, the magnitude of metal exposure, and the presence of certain HLA class II alleles. We show that the predictive algorithm developed from this investigation performs to an accuracy suitable for clinical use, with weighted mean survival probability errors of 1.8% and 3.1% for pre-operative and post-operative models respectively.

**Conclusions** The development of DTH following joint replacement appears to be determined by the interaction between implant wear and a patient's genotype. The algorithm described in this paper may improve implant selection and help direct patient surveillance following surgery. Further consideration should be given towards understanding patient-specific responses to different biomaterials.

## Plain language summary

Millions of joint replacement surgeries are carried out across the world annually. In this surgery, the joint is replaced with an artificial implant. Most implants are made of cobalt chrome (CoCr). Some patients develop allergic responses to these implants, resulting in pain and tissue damage and repeat surgery. We identified patients who had developed allergies to their CoCr hip implants and compared their genes to those of patients who remained symptom-free. Having identified genes that increased the likelihood of a patient developing an allergic response, we invited additional patients to contribute samples for gene testing. Using the combined data, we used a computer algorithm to predict allergic responses based on a patient's genes, age, and gender. The algorithm performed with sufficient accuracy to be usable in clinical practice to guide implant selection preoperatively and guide patient follow-up post-surgery.

[1] ExplantLab, The Biosphere, Newcastle Helix, Newcastle-upon-Tyne, England. [2] Newcastle University, Newcastle-upon-Tyne, England. [3] Yale-NUS College, 16 College Avenue West, 138527 Singapore, Singapore. [4] Department of Medical Genetics, University of Oslo and Oslo University Hospital, Oslo, Norway. [5] Royal Perth Hospital, Perth, Australia. [6] School of Medicine, University of St Andrews, St Andrews, Scotland. [7] University Hospital of North Tees, Stockton, England. [8] Hospital for Special Surgery, New York, USA. ✉email: davidl@explantlab.com

Hip joint replacement surgery (hip arthroplasty) has proven to be extremely successful in the treatment of end-stage hip arthritis. As a result, there are now approximately 2 million hip arthroplasties carried out in countries of the Organisation for Economic Co-operation and Development (OECD) alone[1].

Conventional total hip replacements (THRs) are composed of a metal femoral head which articulates against a polyethylene (plastic) cup or liner[2]. The lifespan of these so-called metal on polyethylene (MoP) prostheses may be limited in younger, more active patients. This is because during activities of daily living, the harder metal head wears away the plastic component. The release of greater amounts of wear debris over time increases the probability of a macrophage-driven, adverse immune response developing in the periprosthetic tissue[2]. The result of this is wear-induced osteolysis, in which the bony architecture surrounding the implant becomes compromised and the component/components loosen[3]. In this situation, revision surgery must be undertaken and a new device implanted.

Metal on metal (MoM) hip resurfacing prostheses were reintroduced at the turn of the century to address this problem[4]. In hip resurfacing surgery, the damaged articular surface is removed from the native femoral head and is replaced by a hollow CoCr femoral component, which articulates against a CoCr acetabular component. It was hoped that removal of the softer polyethylene from the bearing combination would lead to a reduction in wear debris and, therefore, an increase in implant longevity.

The initial early success of the Birmingham Hip Resurfacing (BHR) in young males[4], saw a rapid expansion in the market, with ever-widening patient eligibility criteria and several new prostheses released from competing manufacturers[5]. The technology was then adapted for use in THRs, so that patients without sufficient bone quality to accommodate a resurfacing might benefit from the perceived advantages of decreased wear and increased stability afforded by the large diameter metal bearings[6].

From 2005, there began to emerge an increasing number of case reports which described MoM hip patients returning to clinic with delayed onset groin pain[7]. At revision surgery, large, sterile fluid collections were encountered in the joint capsule[8]. Histopathological examination of excised periprosthetic tissues identified a macrophagic infiltration - as previously encountered in MoP prostheses - but frequently the macrophage response was accompanied by a perivascular T lymphocyte infiltrate. In the most severe cases, the perivascular cuffs had expanded in circumference to coalesce, leading to the formation of germinal centres and destruction of the synovial surface[9]. In a seminal paper, Willert et al. coined the term aseptic lymphocyte-dominated vasculitis association lesion (ALVAL) to describe these histopathological features[10]. ALVAL can frequently be associated with the destruction of local tissues including bone, muscle and neurovascular structures[11]. The lesions are progressive, and if revision surgery is delayed, the incidence of major post operative complications increases[12]. Reimplantation with CoCr components may lead to rapid recurrence of symptoms[13]. The overall clinical and histopathological picture is consistent with a delayed-type hypersensitivity (DTH) response to the metal debris shed from the prostheses[10]. Initially thought to be an idiopathic, rare phenomenon, failure of MoM THRs secondary to ALVAL has been reported to reach over 30% at six years[14].

Note: In this paper we use the terms DTH and ALVAL interchangeably, with ALVAL the preferred term when referring specifically to MoM hip patients in the current study.

Studies have shown that the risk of tissue damage is increased when prostheses shed greater volumes of metal debris[15]. Consequently, in 2012, the Medicines and Healthcare Products Regulatory Agency (MHRA, UK) issued an alert regarding the management of patients with MoM implants. In it they recommended the monitoring of metal concentrations in blood, establishing a threshold of 7 micrograms per litre (μg/l) of Co or Cr as an indicator of an adverse tissue reaction. Although these guidelines were based on a small study involving only 26 patients[16], the guidance has not been substantially modified since its first release. (http://www.mhra.gov.uk/home/groups/dts_bs/documents/medicaldevicealert/con155767.pdf) However, patients display varying tolerances to metallic debris[17], with female patients apparently at greater risk of developing hypersensitivity[18]. There is also evidence to indicate that debris release from the taper junction of THRs exhibits greater immunogenicity than bearing surface debris[19]. This is reflected in the greater failure rates of MoM THRs which has led to their withdrawal from clinical use.

There are two phases of DTH: sensitisation and elicitation. During the sensitisation phase, antigen-presenting cells (APCs) take up, process and display an antigen. APCs migrate to regional lymph nodes where the displayed antigen may activate T4 cells and the production of memory T cells, which migrate to the original site. In the elicitation phase, a subsequent exposure to the antigen leads to its re-presentation to memory T cells with the release of T cell chemokines and cytokines such as interferon-gamma, which enhance the inflammatory response.

A critical factor in the development of DTH, therefore, is the presentation of a specific peptide/antigen at the peptide binding groove of an APC; a competitive process.

Metals are capable of provoking a variety of T cell-mediated, HLA-linked diseases, such as chronic beryllium disease[20], Co hard metal lung disease[21], and contact hypersensitivities[22]. Three pathogenic mechanisms have been described: self peptides held in the binding groove of an MHC molecule form complexes with metal ions, with the resulting complexes acting as antigens[23]; T cells recognize metal-induced changes to the MHC molecule itself[24]; metals directly affect the processing of self-peptides, resulting in T cells reactive to cryptic self-peptides[25].

Which mechanism may be the most important in the initiation of the ALVAL response? Previous research and clinical experience indicated that the first mechanism was the most likely, and that the N terminal sequence (NTS) of albumin was the prime candidate peptide sequence to investigate[26]. We, therefore, hypothesized that individuals developing ALVAL may have greater frequencies of (HLA gene encoded) peptide binding grooves with greater affinities for the NTS of albumin.

In this investigation, we demonstrate that variation in HLA class II genotype influences an individual's susceptibility to DTH following implantation with a CoCr hip prosthesis. We go on to describe the development and validation of a machine learning algorithm to investigate the possibility that a patient's genotype and basic clinical parameters may be used to predict the development of DTH.

## Methods

**Patients and hospital centres.** Following Health Research Authority ethical approval (IRAS reference 227785), the study commenced at a single centre (centre 1, United Kingdom) where a large number of MoM hip arthroplasties were performed between 2002 and 2010. These patient cohorts have been described in full in previous publications[27]. The patients have been kept under surveillance with annual clinical review and blood metal ion testing. As part of an ethically approved project (IRAS reference 14119), patients who undergo revision of their MoM hip prostheses have: undergone metal ion testing to assess Co and Cr concentrations in their blood, serum, and hip joint synovial fluid samples; their explanted prostheses analysed to

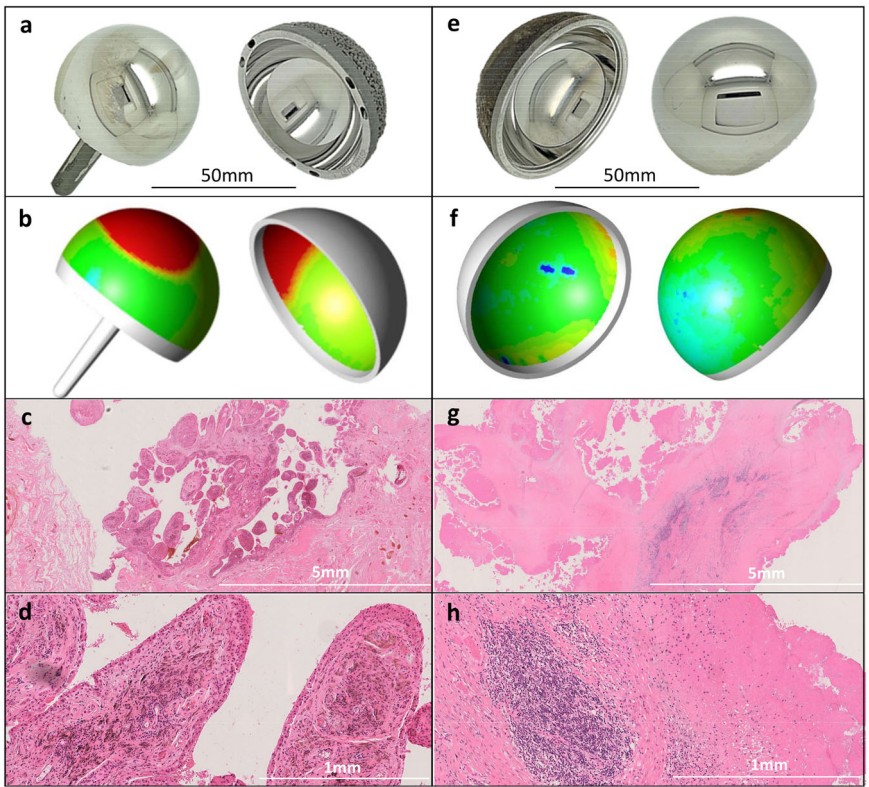

**Fig. 1 Explant analysis and tissue responses.** MoM hip components are manufactured from standard, medical-grade CoCr alloy (ASTM-75 or ASTM-1537), which is composed of approximately 65% Co and 30% Cr by weight. Explanted prostheses can be reverse engineered using coordinate measuring machines (CMMs) to quantify the volumetric material loss through wear and corrosion (shown in red in the wear maps below). Serum or whole blood Co and Cr concentrations provide a reliable in-vivo surrogate measure of the rate of this material loss[80]. Panels a,b,c and d relate to the same patient, whose blood Co concentration was elevated at 20.1 µg/l just prior to revision. **a** The explanted 46 mm diameter Birmingham Hip Resurfacing. **b** The corresponding CMM generated wear map (volumetric wear rate of 25mm$^3$/year). **c** and **d** The synovial tissue sections (hematoxylin and eosin (H&E) stained), (X2 magnification and x15 magnification respectively) showing heavy macrophage infiltration, no lymphocytes. Panels **e–h** relate to a second patient, whose blood Co concentration was 1.5 µg/l just prior to removal of her ASR XL THR. **e** The explanted 45 mm diameter prosthesis. **f** The corresponding CMM generated wear map (volumetric wear rate of 1.5mm$^3$/year). **g** and **h** The synovial tissue sections (H&E stained), (x2 magnification and x15 magnification respectively) showing a heavy perivascular lymphocyte infiltrate and extensive synovial necrosis (severe ALVAL).

determine their volumetric wear; and tissue samples excised at revision surgery assessed by a specialist histopathologist (SN) (Fig. 1). The total number of revision cases in the database at commencement of the current study was 420. All patients included in the study gave informed consent.

**Blood/serum metal ion testing.** We have carried out a substantial amount of work detailing the relationships between volumetric wear of implants and the corresponding concentrations of Co and Cr ion in the blood, serum, and synovial fluid fractions[28]. Samples were tested using the generally accepted method of inductively coupled plasma mass spectrometry (ICP-MS) at accredited laboratories[19,29].

*Wear analysis.* Explanted prostheses were analysed using a coordinate measuring machine (Legex 322; Mitutoyo Ltd, Halifax, United Kingdom) to calculate the total amount of material that had been lost from the components in vivo: 'total volumetric wear', measured in mm.$^3$ The total volumetric wear was divided by the number of years in vivo to calculate a mean 'volumetric wear rate' (expressed in mm$^3$/year) which was the value used in the statistical analyses. The accuracy of the volumetric wear analysis performed on these types of explanted components has been validated and is of the order of 0.5 mm$^3$ for a bearing surface and 0.2 mm$^3$ for a female taper surface[30]. In this paper, wear rates

refer only to CoCr material loss. For resurfacings, therefore, the wear rates refer only to the bearing surface wear rates (combined femoral head and acetabular component volumetric wear rates). For THAs, 'total volumetric wear rates' include the bearing wear as well as the wear from the female taper surface (Fig. 2). THAs in the study were used with titanium stems. We have previously demonstrated that titanium release is small in comparison with CoCr[31].

*Histopathological tissue assessment.* This was carried out as has previously been described in greater detail[32]. Samples were taken from between two and four periprosthetic sites. Up to ten paraffin blocks were processed per site. Samples were also sent for microbiological testing to exclude sepsis. A single consultant histopathologist (SN) examined the slides independently of the clinical findings, blinded to the results of the wear or metal ion analyses. Note: Adverse reaction to metal debris (ARMD) is an umbrella term which refers to clinical signs and symptoms association with metal debris exposure[33]. The typical immunological response to metal debris is limited to a macrophage infiltrate[34]. ALVAL is a subset of ARMD, referring to the additional lymphocyte infiltrate and histological features of DTH. The hallmark of ALVAL/DTH is the development of a perivascular lymphocytic cuffs which increase in thickness as the recruitment of lymphocytes is further stimulated. In more severe cases, these cuffs can expand to develop into aggregates or coalesce into one

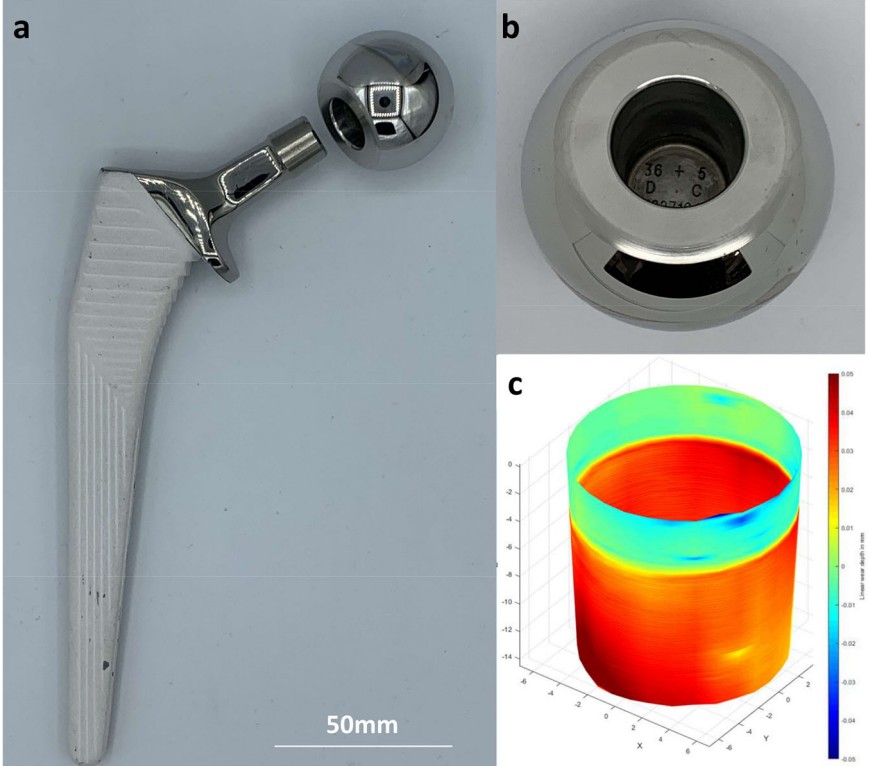

**Fig. 2 The structure of total hip replacements (THRs). a** In total hip arthroplasty, the femoral neck is sectioned and a femoral stem (titanium, uncemented for the patients in this study) placed down the femoral canal. The femoral head is press fit on to the stem, creating the taper junction. **b** Metal debris can be generated from the taper junction, the great majority of which is released from the CoCr head. For the explanted THRs in this study, material loss from the bearing and tapers was quantified using a coordinate measuring machine. **c** A taper wear map is shown, with red areas indicating areas of material loss greater than 50 microns in depth.

another, forming larger aggregates. These higher-grade ALVAL responses are associated with the development of tertiary lymphoid organs in the local tissue. As part of routine clinical practise, the ALVAL response in the tissue samples in this study was graded from 0 (absent) to 3 (severe) according to the integrity of the synovial membrane and the extent of lymphocytic infiltration (Fig. 1), a classification system which has shown good intra and interobserver reliability[32].

**Investigation of genetic associations using extreme phenotype group comparison.** From the hospital database, we identified four groups of patients, to represent the different phenotypes: *patients with joint failure* who developed moderate/severe ALVAL in association with prostheses wearing at lower than the median wear rate of the total revision cohort; *patients with joint failure* who developed moderate/severe ALVAL in association with prostheses wearing at greater than the median wear rate of the total revision cohort; *patients with joint failure* with a pathological response limited to macrophage infiltration, no lymphocyte infiltration identified; *patients with joints remaining in situ* who were pain-free and satisfied with the results of their hip arthroplasties at a minimum of ten years post surgery. We wrote to these patients explaining the nature of the study and invited them to submit a sample for DNA analysis.

*DNA sample collection and processing.* A combination of ORA-collect OCR-100 buccal swabs and Oragene DNA OG-610 saliva collection kits (both DNA Genotek Inc, Ontario, Canada) were used to collect samples for DNA extraction. DNA was extracted using a Roche MagnaPure Compact automated platform (Roche Holding AG, Switzerland). DNA was then quantified using a

Thermo Fisher Qubit dsDNA BR Assay kit (Thermo Fisher, Massachusetts, United States) with standardisation to 25 ng/µl. HLA genotyping was then performed using One Lambda AllType NGS kits (One Lambda, USA), with the Illumina MiSeq platform (Illumina, USA). Full gene sequencing was carried out for HLA-A, -B, -C, -DQA1 and -DPA1, and partial gene sequencing for HLA-DRB1, -DRB345, -DQB1 and -DPB1 (with omission of exon 1). HLA genotypes were analysed using One Lambda TypeStream Visual 1.3 software (One Lambda, USA).

Global locus-wise association for each HLA gene was performed using UNPHASED v 3.0.13. Haplotypes were estimated for DRB1-DQA1-DQB1 also in UNPHASED[35], and then the distribution of the HLA class I and II alleles were compared between groups using a standard approach[36]. The genotypes for each HLA gene were transformed into dosages of each individual allele from the patient population, where 2 denoted two copies of an allele, 1 denoted one, and 0 denoted zero copies. These values were then entered as predictor variables in a logistic regression analysis. Multiple models were tested, comparing the extreme phenotype groups described above, and these were also compared to a background population from the United Kingdom. All models were also tested with sex as an additional covariate and also age plus sex as covariates.

**In silico analysis of peptide-HLA class II binding affinity and Cox proportional hazards modelling**
*Peptide binding analysis.* We used validated software to model the peptide-binding grooves encoded by an individual's HLA genotype and to determine the resulting binding affinity between these binding grooves and an array of naturally occurring peptides[37]. Using this approach, we sought to: identify HLA genes associated

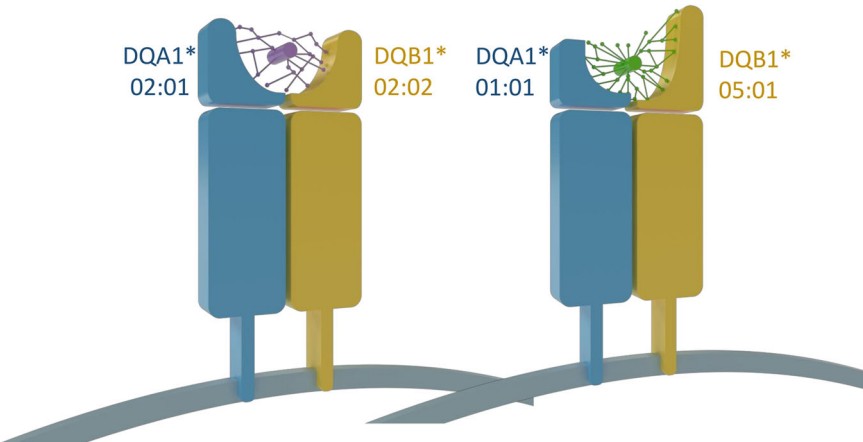

**Fig. 3 MHC structures and peptide presentation.** The HLA-DQ molecule is an αβ heterodimer of MHC class type II. The three-dimensional shape of the peptide-binding groove is formed by the combination of α and β chains, which are genetically encoded by the HLA-DQA1 and HLA-DQB1 alleles respectively[81]. The structure of the peptide-binding groove determines which peptides (foreign or self) are presented at the cell's surface[81], as is shown in the schematic. As an example, in coeliac disease, patients possess HLA alleles that encode for peptide binding grooves with structures suited to the presentation of gluten-derived peptides[82].

with the development of ALVAL; determine whether HLA genes are associated with the development of ALVAL at low rates of wear encode for peptide binding grooves with higher affinities for the N terminal metal-binding sites of albumin.

All HLA-DQA1, -DQB1, and DRB1 alleles were selected to assess the peptide binding affinity of their corresponding peptide-binding proteins. HLA-DR is represented by HLA-DRA/DRB1 dimer. Since HLA-DRA is considered monomorphic, we only used HLA-DRB1. HLA-DQ is represented by the HLA-DQA1/DQB1 dimer. A basic schematic of the HLA-DQ structure and how it relates to peptide binding is shown in Fig. 3.

FASTA-formatted protein sequence data were retrieved from the UniProt database (www.uniprot.org) for human serum albumin (P02768). We extracted the first 15 amino acids of the N terminal (DAHKSEVAHRFKDLG), a sequence which includes two recognised Co binding sites. Predictions for HLA binding to this sequence were performed using NetMHCIIpan4.0[37]. The rank binding affinities were calculated for all the possible DQ and DRB1 combinations. We used the %EL rank score as the primary binding metric, as advised by the software developers[38]. We investigated whether the binding scores influenced the risk of developing ALVAL over time using Cox proportional hazards modelling. Multiple survival models were constructed to explain the development of time dependent prosthetic failure associated with mild/moderate or severe ALVAL, using the following independent variables: NTS binding affinity; pre-revision blood Co concentrations; pre-revision blood Cr concentrations; patient sex; patient age at the time of primary surgery; the presence of bilateral prostheses; type of prosthesis (THR versus resurfacing arthroplasty).

**Expansion of data set, the inclusion of patients from other centres and development of machine learning algorithm.** We then invited all remaining patients in the database who had undergone revision surgery for whom there was a full complement of clinical data, including explanted components available for analysis. We also invited all remaining patients under regular follow up who were recorded to be asymptomatic at greater than ten years follow up. Concurrently, we expanded the study to include two other units. Centre 2 is a major specialist orthopaedic unit in New York, United States. Centre 3 is a teaching hospital and tertiary referral centre in Western Australia. The units

manage the follow up of MoM patients in a similar way and also routinely carry out analysis of explanted components. A similar research protocol was followed, with patients who were asymptomatic as well as those who had experienced failure of their joints invited to give a sample for DNA analysis. Relevant national and local ethical approvals were sought and granted (Protocol 2020-208, IRB approval for the United States; study RGS0000003851 Human Research Ethics Committee approval for Australia). The same parameters were recorded as at centre 1, with all patients giving informed consent. When all samples had been analysed, the data set was randomly split 70/30, with the larger set used to train a machine learning algorithm for the prediction of the development of ALVAL. The remaining data was held back, blinded from the analysts and used to test the algorithm when it was finalised.

Two models were trained to predict hazard ratios and survival functions up to ten years after implantation of a MoM prosthesis for pre-operative and post-operative patients. The first was a model to preoperatively predict the development of ALVAL. For this model, metal exposure was divided into two groups: low wear (Co concentrations stabilise to <2 μg/l) and increased wear (Co concentrations stabilise to ≥2 and ≤4 μg/l). 4 μg/l equates to approximately three times the wear rate of a well-functioning device. It was therefore not felt necessary to provide a preoperative prediction for metal concentrations above this level. A second model was developed to predict the development of ALVAL in the post-operative period, in which actual measured Co and Cr concentrations could be used in the modelling.

*Statistics and machine learning approach.* As the training and test set are assumed to be drawn from the same probability distribution, they should be identically distributed[39]. We, therefore, formulated our test set by randomly sampling the full dataset (without replacement) stratified on the event indicator. The training data was composed of the remaining samples.

Feature engineering was carried out on the training data to identify features that best predicted risk of failure due to ARMD and ALVAL within ten years of implantation of a MoM prosthesis. Boruta[40], a random forest feature selection algorithm was applied to 2939 features, generated from a combination of: patient features; binding affinities of *cis* and *trans* haplotypes; binary presence of *cis* and *trans* haplotypes; *cis* and *trans* haplotype gene dosage; thresholding binding affinities of *cis* and

**Table 1 Clinical details of all the patients from all centres in the study, divided by clinical outcome.**

|  | Total | Asymptomatic | Failed |
|---|---|---|---|
| Total number of patients | 606 | 430 | 176 |
| Total number of hips | 711 | 535 | 176 |
| Follow up (years) | 10 (1–20) | 12 (3–20) | 6 (1–15) |
| Age (range) | 55 (25–85) | 54 (25–78) | 58 (25–85) |
| % male patients | 66% (397:209) | 74% (320:110) | 44% (77:99) |
| Resurfacings vs THRs | 468 vs 138 (77%) | 43 vs 430 (90%) | 46% (81:75) |
| % patients with bilateral prostheses | 24% | 24% | 24% |
| BMI | 26.6 | 26.7 | 26.3 |
| Median (range) Co (µg/l) | 2.00 (0.1–271.0) | 1.50 (0.1–34.4) | 7.60 (0.7–271.0) |
| Median (range) Cr (µg/l) | 2.50 (0.2–108.4) | 2.00 (0.2–18.6) | 7.01 (0.7–108.4) |

*trans* haplotypes to generate categorical features; polynomial and interaction features. The algorithm removed features that were identified as being less relevant than random features in an iterative supervised fashion to avoid overfitting. Features that were identified as being associated with ALVAL were used to train gradient boosted survival analysis machine learning models with a Cox proportional hazards loss function and a regression tree base learner[41,42]. Regularisation was employed to reduce overfitting on the training data. Nested 5-fold cross-validation (CV) was used on the training data to enable better estimation of generalisation error and reduce model selection bias[43,44]. Hyperparameters of both models were optimised using a successive halving random search[45,46]. Integrated Brier Score (IBS) was chosen as the scoring function. Cross-validated probability calibration did not yield improvements in IBS and Integrated Calibration Index (ICI).

After selecting the best-performing model based on the IBS assessed on the training data, the model was then used to predict on the test set. IBS, Uno's c-index[47], time-dependent AUROC (ROC(t))[48], and ICI performance statistics were computed[49]. We used Austin et al.'s adaptation of ICI for survival analysis problems[50]. Confidence intervals were estimated using the Bootstrap method. After completion of performance evaluation, each model was refit on the training and test data and hyperparameters were returned. The models were then serialised and integrated into a cloud-hosted pipeline for inference via a web app.

**Reporting summary**. Further information on research design is available in the Nature Research Reporting Summary linked to this article.

## Results

**Investigation of genetic associations through comparison of extreme phenotype groups (unit 1).** There was a response rate of around 60% in each of the four patient groups, resulting in a total of 161 patients who gave saliva or buccal samples for DNA analysis. There were no significant differences in age or sex between responders and non-responders (chi-squared test, $p = 0.548$). Patient details and allele frequencies between groups can be seen in Supplementary Table 1 and Supplementary Data File 1. There was a bias towards female sex, increased age, and THRs in patients developing ALVAL in response to lower wear. The strongest signals were found with two haplotypes, which had opposing associations with ALVAL. The dominant, significant positive association was seen with DQA1*02:01, DQB1*02:02, and DRB1*07:01. These alleles were increased across all phenotypic subtypes in patients with prosthetic failure. The alleles were in strong linkage disequilibrium and occurred on one associated haplotype. A protective effect was seen with the alleles DQA1*01:01, DQB1*05:01, and DRB1*01:01. These alleles, also

in strong linkage disequilibrium and occurring on one associated haplotype, were found in significantly higher frequencies in patients without ALVAL. Class, I HLA allele distributions did not differ between the groups.

**In silico analysis of peptide-HLA class II binding affinity and Cox proportional hazards modelling (unit 1).** DQA1*02:01-DQB1*02:02, the haplotype with the strongest positive association to ALVAL, exhibited the strongest binding affinity to the NTS of albumin. DQA1*01:01-DQB1*05:01, the commonly occurring haplotype with the strongest negative association with ALVAL, exhibited one of the weakest binding affinities in the dataset (rank 15 out of 17 haplotypes). Cox proportional hazards modelling, incorporating NTS binding affinities as a continuous measure of DQ haplotype, demonstrated that pre-revision blood Co and Cr concentrations, female sex, and greater NTS binding affinity were significantly associated with increased risk of early ALVAL-associated failure. These models were consistent using different thresholds of ALVAL (mild and above versus moderate and above (Supplementary Table 2). No relationship was identified between prosthetic failure and the binding affinity values derived from DRB1 molecules. Therefore, a decision was made to expand patient recruitment but focusing solely on DQ molecules.

**Expansion of data set, recruitment of patients from other centres, and development and testing of a machine learning algorithm (units 1, 2, and 3).** A total of 606 DNA samples, from 397 males and 209 female patients, were successfully typed. This included 320 patients from the United Kingdom, 259 from the United States, and 27 from Australia. Patient demographics and clinical parameters can be seen in Tables 1 and 2 and Supplementary Table 3.

The clinical details of the training and validation sets are shown in Supplementary Table 4. Supplementary Table 5 shows the results of performance evaluation of the presented models on the test set. Taper-dominated wearing THRs were excluded from the test set for the ALVAL pre-operative model to better fit the clinical context which this model would be exposed to (very few, if any, MoM THRs are currently implanted; only resurfacings).

We opted to use Uno's variation of c-index for measuring discriminatory performance, which addresses the overly optimistic results observed for Harrell's c-index with increasing censoring frequency. Whilst AUROC is equivalent to c-index for binary outcomes, ROC(t) provides a measure of performance for a given time of interest. We, therefore, used an ROC(t) measure which accounts for censored patients using the Kaplan-Meier estimator to assess discriminatory performance at discrete time periods. Integrated Brier Score (IBS) was chosen as the scoring function as our model's primary use was predicting

**Table 2 Cox proportional hazards modelling, all 606 patients involved in the study included.**

| Variable | Coeff | Standard error | P-value | Hazard ratio (HR) | HR Lower CI (95%) | HR Upper CI (95%) |
|---|---|---|---|---|---|---|
| **Model 1: Survival based on ALVAL severity of mild, moderate, or severe** | | | | | | |
| Rank binding affinity for NTS | −1.463 | 0.404 | <0.001 | 0.232 | 0.105 | 0.511 |
| Log normalised cobalt concentration | 1.649 | 0.136 | <0.001 | 5.202 | 3.982 | 6.797 |
| Age | 0.005 | 0.011 | 0.667 | 1.005 | 0.984 | 1.026 |
| Sex-M | −0.571 | 0.190 | 0.003 | 0.565 | 0.389 | 0.820 |
| Type-THR | 0.779 | 0.201 | <0.001 | 2.180 | 1.471 | 3.230 |
| **Model 2: Survival based on ALVAL severity of moderate or severe** | | | | | | |
| Rank binding affinity for NTS | −2.532 | 0.544 | <0.001 | 0.079 | 0.027 | 0.231 |
| Log normalised cobalt concentration | 1.656 | 0.178 | <0.001 | 5.236 | 3.696 | 7.417 |
| Age | 0.013 | 0.014 | 0.359 | 1.013 | 0.986 | 1.041 |
| Sex-M | −0.631 | 0.247 | 0.011 | 0.532 | 0.328 | 0.863 |
| Type-THR | 0.728 | 0.264 | 0.006 | 2.070 | 1.235 | 3.470 |

Patients were censored based on a minimum ALVAL grade of "mild" and above, then a second model was constructed with patients censored only if they had "moderate" ALVAL and above.

hazard ratios and survival curves and therefore discriminatory ability and calibration were equally important.

For all models, the c-index and ROC(t) scores suggested a high degree of discrimination, whilst the IBS indicated good calibration and further backed up the indication of high discriminatory ability. The ICI scores supported the indication of good calibration and showed that at ten years, the weighted mean survival probability error was 1.8% and 3.1% for pre-operative and post-operative ALVAL models respectively (Supplementary Table 5). Supplementary Figs. 1 and 2 show ALVAL ROC(t) for pre-operative and post-operative models from two to ten years after implantation. The ALVAL pre-operative model peaked in performance at two years after which a similar performance was observed from three to ten years. Similarly consistent performances were observed for the ALVAL post-operative model.

**Survival analysis using total data set.** Kaplan Meier and Cox proportional survival analyses involving all patients confirmed the initial single centre results, showing that greater Co (and Cr as the rank correlation between the two elements = 0.816, $p < 0.001$) concentrations, female sex, THR prostheses, and genotypes with greater NTS binding affinities were significantly associated with greater risk of ALVAL related prosthetic failure (Table 2 and Figs. 4, 5 and 6).

## Discussion

At the height of its popularity around 15 years ago, MoM technology was used in around 30% of all hip replacements implanted in the United States[51], and in total, over 1 million MoM hips have been implanted across the world. Due to high complication rates, the use of MoM bearings has dramatically declined, and is now restricted to hip resurfacing procedures carried out in a limited number of centres[52].

Research carried out over the last three decades—in the fields of dermatology and respiratory medicine - has identified HLA genes as key factors in the development of metal sensitivity. Orthopaedic researchers have also identified links between HLA genes and adverse local tissue responses, but, to date, the published studies have involved limited numbers of patients[53,54]. Our results indicate that the development of DTH/ALVAL following joint replacement is determined by an interaction between patient sex, genotype, and the volume of metal debris generated from a prosthesis. In present-day clinical practise, genetic predisposition to DTH is not routinely considered nor tested for in the selection of an orthopaedic implant. Some centres carry out investigations such as skin patch testing or perform lymphocyte proliferation assays to identify patients who report a metal allergy pre-operatively. However, these tests have faced continued scrutiny as to their accuracy and true clinical relevance[55,56]. In this paper we have described the development and validation of an algorithm which may help identify patients at greater risk of DTH in order to guide implant selection and inform post-operative surveillance.

**Development of ALVAL is associated with HLA genotype.** In the initial part of this investigation, we showed that patients developing ALVAL to low wearing prostheses possess different frequencies of specific HLA-DQ haplotypes when compared to those who remain asymptomatic at long-term follow-up. Validated software enables the virtual construction of peptide binding grooves encoded by an individual's HLA genotype[37]. This allows the calculation of the binding affinity between a particular HLA encoded peptide-binding groove and an array of naturally occurring peptides. We used this software to demonstrate that HLA-DQ haplotypes encoding for peptide binding grooves with greater affinity for the NTS of albumin present a higher risk of ALVAL.

**Albumin and metal binding.** The NTS of albumin contains two metal ion binding sites. The first arises from the first triplet amino acid motif of albumin: Asp1–Ala2–His3 (the N terminal site)[57]. The second (also termed site B), is partially composed of His9 and Asp13[58]. While site B exhibits greater binding affinity for Co, the N terminal site is formed from a continuous amino acid sequence, not reliant on connections between domains to maintain its metal-binding properties. It is thus more likely to remain intact through cellular processing and presentation[57]. Under normal circumstances, the N terminal site is generally occupied by nickel or copper[59]. However, this site is also recognised to bind Co ions, particularly when there are changes in the relative concentrations of metal in the surrounding fluid. This is indeed the case in patients suffering DTH reactions, whose joints often develop large, albumin-rich synovial fluid collections containing massive concentrations of Co. Studies have shown that in these fluid collections, Co is almost exclusively bound to albumin[19,60].

**Although patients display varying susceptibility, most patients require exposure to elevated wear rates to develop ALVAL.** Metal ions can form complexes with self-proteins held at the

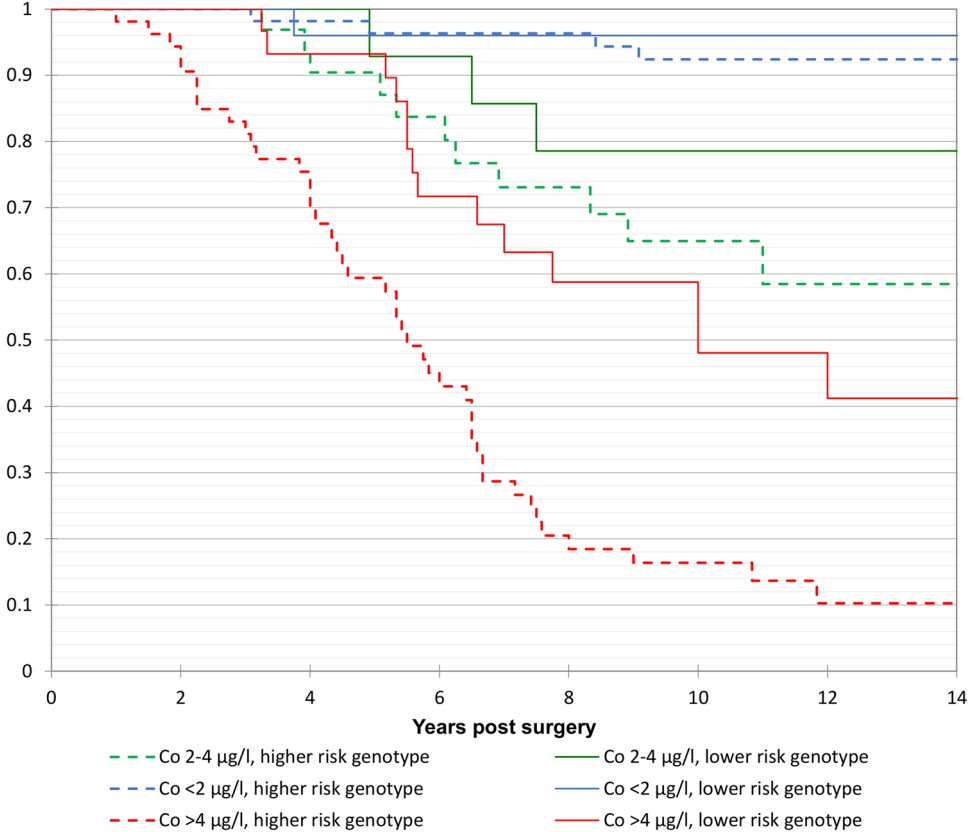

**Fig. 4 Kaplan–Meier survival analysis including all female patients in the study.** Failure secondary to ALVAL (mild/moderate/severe). The patients have been sub-grouped according to their HLA genotypes and blood Co concentration. Higher risk genotype defined by the top 50% NTS binders, lower risk by the lowest 50%. Cox proportional hazards modelling using "Co < 2, lower risk genotype" as the reference group ($n = 55$) returned hazard ratios (HR)(95% CI) of 1.95 (0.22–17.0) for Co < 2, higher risk genotype ($n = 25$); 5.95 (0.62–57.2) for Co 2–4, lower risk genotype ($n = 32$); 12.54 (1.62–97.2) for Co 2–4, higher risk genotype ($n = 14$); 18.7 (2.45–142) for Co>4, lower risk genotype ($n = 53$); 53.1 (7.26–388) for Co > 4, higher risk genotype ($n = 30$).

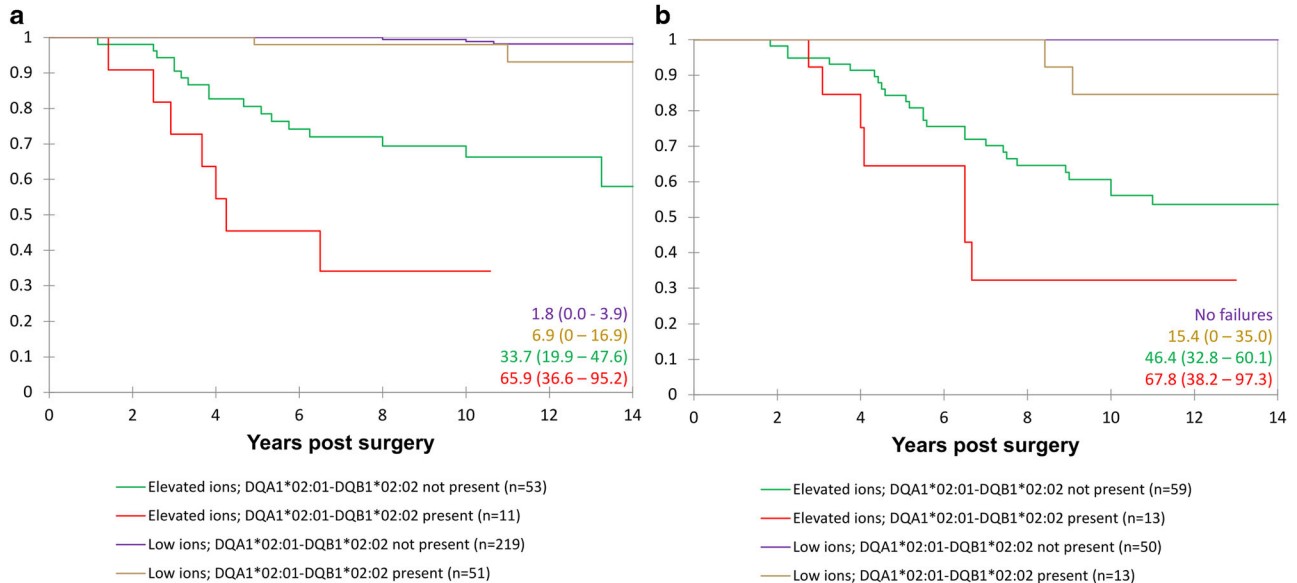

**Fig. 5 Kaplan–Meier survival analysis including all patients in the study. a** Male patients. **b** female patients. Failure defined as revision secondary to ALVAL (moderate or severe). The patients have been sub-divided according to implant type (hip resurfacing or THR) and genotype (possession of HLA-DQA1*02:01-DQB1*02:02). Failure rates at 12 years are shown at the bottom right of the charts (with 95% confidence intervals).

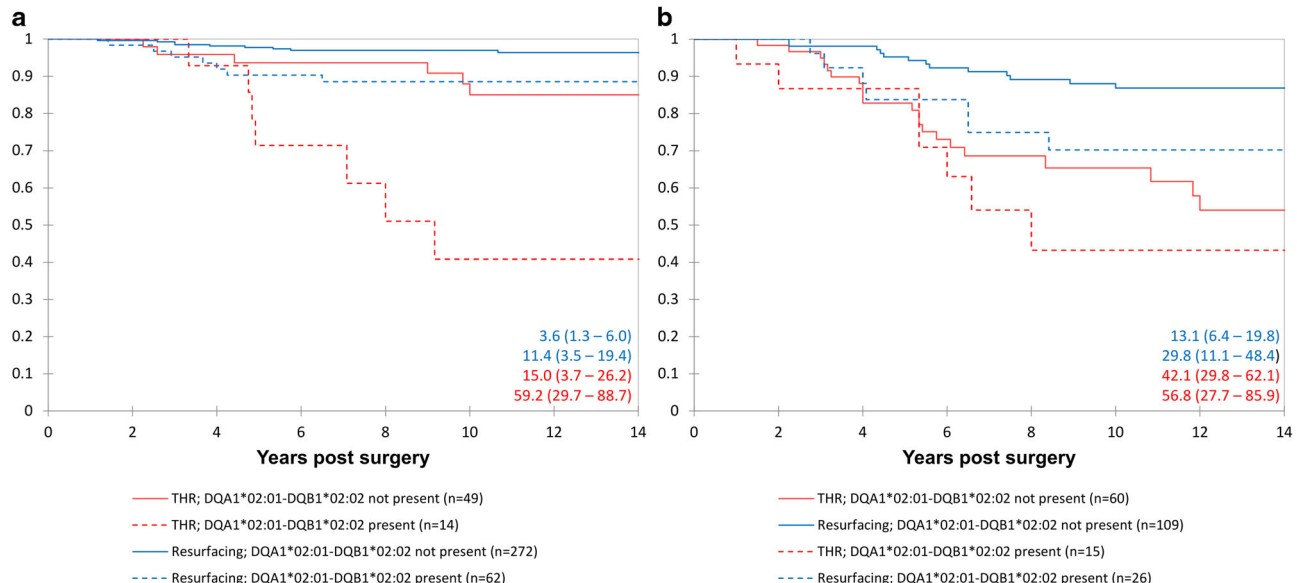

**Fig. 6 Kaplan Meier survival analysis of the hip resurfacing patients. a** Male patients. **b** female patients. Failure defined as revision secondary to ALVAL (mild, moderate, or severe). Patients were subdivided according to blood ion concentrations: ("low ions" (Co < 2; Cr < 4) and "elevated ions" (Co ≤ 2 or Cr ≤ 4 μg/l)) and genotype (presence or absence of DQA1*02:01-DQB1*02:02). Failure rates at 12 years are shown on the bottom right of the charts (with 95% confidence intervals).

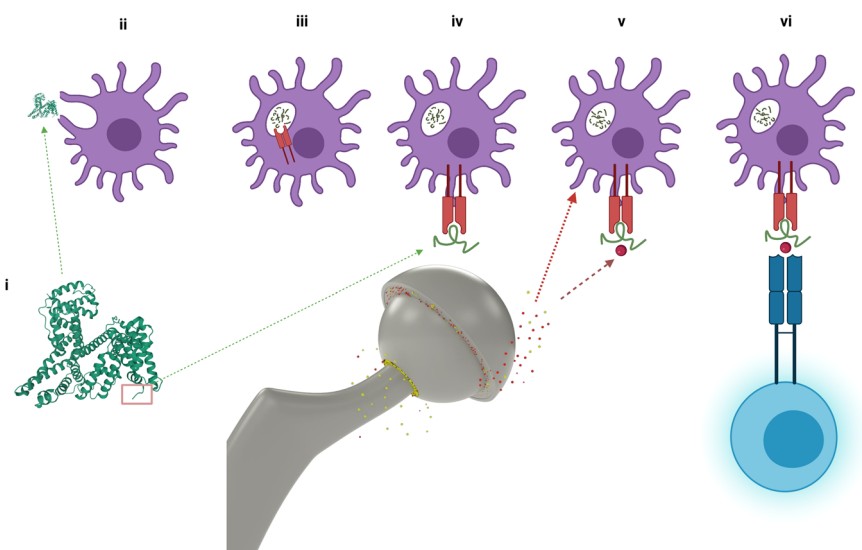

**Fig. 7 Pathogenesis of DTH.** (i) The N terminal sequence (NTS) of albumin (red rectangle) contains two recognised metal-binding sites. (ii) Endogenous (self-proteins, including albumin) and exogenous (pathogens/ingested substances) proteins are ingested by APCs, which include macrophages and dendritic cells (purple). (iii) The ingested proteins are acidified in the lysosomes of the APCs(29), where peptidases act to section the original proteins into their shorter, constituent peptides. These peptides compete for the binding groove of class II MHC complexes (red). (iv) If a peptide forms a stable complex with the MHC molecule, the resulting MHC:peptide complex is transported to the cell surface (v) Self-peptides held in the binding groove of an MHC molecule can form complexes with metal ions (dark red sphere) released from implant surfaces (broken dark red arrow) to form haptens. (vi) The MHC:peptide:metal complex may activate T4 lymphocytes resulting in sensitisation. This will happen if the lymphocyte has been activated via pattern recognition receptors (PRRs), which can occur due to local cellular damage through metal toxicity (broken red arrow). Figure created with Biorender.com.

peptide-binding groove of APCs; these metal peptide complexes can provoke a response from T4 helper cells. However, the presentation of a peptide:MHC complex by an APC does not automatically lead to sensitisation. This is demonstrated by the low ALVAL rates in our patients who possess higher risk haplotypes but have low blood metal ion concentrations. Sensitisation requires lymphocyte activation, and for a lymphocyte to become activated, the APC itself must be in an activated form. APCs possess innate pattern recognition receptors (PRRs) (Fig. 7),

which, when stimulated, promote activation and migration of APCs from the site of exposure to the draining lymph nodes. This process leads to expansion and survival of metal-reactive memory T cells that circulate throughout the body. Metals can activate PRRs either directly, or indirectly, through the release of reactive oxygen species, the inflammasome pathway[61,62] or via the induction of necrosis and release of alarmins[63]. An elevation in local metal ion concentrations, therefore, can not only raise the probability of metal:peptide neoantigen presentation, it can also

increase the probability of APC activation and thus T cell sensitisation. Furthermore, an increase in the rate of implant wear can lead to the generation of larger particles which may frustrate effective macrophage phagocytosis, resulting in cell damage, the release of lysosomal products, and a local reduction in pH levels[64].

**The N terminal peptide sequence of albumin is recognised to fragment early in the endolysosomal processing pathway.** Albumin peptides, as with other endogenous peptides, are constantly recycled in the body[65]. This recycling commences following pinocytosis or receptor-mediated cellular uptake, when proteins enter the endolysosomal pathway, and are exposed to an increasingly acidic environment (Fig. 7). As the pH drops, peptidases section the ingested proteins into their smaller constituent peptides. Albumin is protected from this endosomal degradation by the binding of neonatal Fc receptor (FcRn), binding which is initiated at pH values below 6.5[66]. However, N terminal albumin sequences 1–24 and 1–26 are some of the first to fragment under mildly acidic conditions[67], a phenomenon that enables them to act as biomarkers in conditions such as graft versus host disease. Therefore, the NTS could detach from albumin via two mechanisms in different locations: in the synovial fluid itself, or in the endolysosomal pathway, where it is a front runner in the competition to bind with MHC II molecules. Once presented at the cellular membrane, NTS peptides would be exposed to metal ions released from the prosthesis, leading to the formation of metal: peptide complexes (Fig. 7).

**Females are more susceptible to DTH.** As is the case with other (largely HLA mediated) autoimmune/autoimmune-like diseases, females develop ALVAL more readily than male patients. We have previously - incorrectly - ascribed this to the tendency for prostheses implanted into females to wear at higher rates[68]. While it is indeed true that MoM hips implanted into females do tend to generate more wear debris, females appear to be more susceptible to ALVAL following exposure to equivalent amounts of metal debris. Accordingly, only females with low wearing prostheses, or genotypes associated with the lowest NTS binding affinity values, were associated with low rates of ALVAL. We are currently investigating the role that other genes and sex hormones may play in this respect.

**MoM THRs carry a higher risk of ALVAL than hip resurfacings.** In this investigation, we have again demonstrated that metal debris release from a THR prosthesis is associated with a greater risk of DTH. We speculate that this may be due to differences in the mechanism of metal release - with corrosion playing a more dominant role - and the production of metal species (such as hexavalent chromium[19]) with a greater capacity to activate PRRs[69]. The literature now conclusively shows that MoM THRs fail at higher rates than hip resurfacings, and as a result, they are essentially no longer used in common practise[6]. However, the early failure to appreciate the important distinction between the performance of MoM THRs and hip resurfacings has, some would argue, led to unjustified concerns over the dangers of hip resurfacing, a procedure which has shown good results in young, active males[6]. It now seems possible that genotyping could be used to further improve the results of resurfacing, a procedure which has the advantage of allowing the retention of the patient's native proximal femur. Despite this, many surgeons would argue that THRs using modern ceramics and highly cross-linked polyethylenes are preferable, given their proven long-term survival rates and very low reported rates of DTH[6].

**Clinical implications beyond MoM hips.** The results have implications for other types of joint replacements. Almost all commonly used total knee replacements (TKRs) include at least one CoCr component, and revision knee prostheses often incorporate a mixed metal modular junction. Yet, while it is now established that CoCr debris released from MoM hips is of great concern - necessitating specific guidance from orthopaedic societies[17] - there is a lack of consensus as to the clinical significance of metal sensitivity in the field of knee surgery[70]. This may be due to a lack of standardisation in terminology, with the ill-defined condition allergy frequently referred to in the literature concerning knee prostheses[71]. It may be due to the pervasive belief that the amount of CoCr released from TKRs is negligible in comparison to that generated from MoM hips, a belief which lies contrary to the findings reported in simulator and retrieval studies[72,73]. In terms of clinical data, blood metal ion studies involving patients with TKRs are few in comparison to those on patients with MoM hips. The few studies that have been published report median Co concentrations ranging from between 0.28 µg/l (in patients with TKRs with titanium tibial trays)[74], to 4.28 in patients with bilateral knees with CoCr trays[75], and up to 8.80 µg/l in patients with unstable components[76]. These ranges extend well beyond the levels which are associated with DTH/ALVAL in MoM hip patients who possess higher risk genotypes. Only recently have researchers performed large, targeted studies focusing on DTH/ALVAL in failed TKRs. Kurmis et al. found a prevalence of pseudotumours or high-grade ALVALs in 7% of their patients with failed TKAs[77]. These findings were substantiated by Crawford et al.[78], who found that 19.1% of their patients who had undergone aseptic revision were found to have perivascular lymphocytic infiltrate on histological analysis. The aforementioned studies also identified a link between the extent of lymphocyte infiltration on the tissue specimens and the pain levels reported by the patients prior to the revision surgery, raising the possibility that DTH may be an under recognised cause of sub-optimal clinical outcomes following joint replacement. It is notable that studies consistently report complaints of chronic pain in approximately 20% of patients following TKA[79]. Our results indicate that at least 10% of individuals of European descent possess HLA genes which may respond unfavourably to relatively low levels of CoCr exposure.

In conclusion, this study provides further evidence that the clinical success of joint replacement surgery is determined by the interaction between implant, surgeon, and patient-specific factors. At present, the arthroplasty community appears focused on controlling surgical factors, such as improving implant position through the use of robots. We suggest that more resources should be directed towards improving the understanding of host-specific responses to implant materials.

## Data availability
Raw genetic data of the extreme phenotype groups are included in Supplementary Data 1 and Supplementary Table 1. Supplementary Data 2 provides the source data for Figs. 4, 5, and 6. Further demographic and clinical details of study patients are included in Supplementary Tables 3 and 4. Patient consent was not obtained to share the individual genetic results on a public repository. However, further data that support the findings of this study are available from the corresponding author upon reasonable request following ethical approval.

## Code availability
The computer code software algorithm is proprietary and therefore not available to the general reader. The code can be supplied from the corresponding author upon reasonable request for use in non commercial, ethically approved research

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

## Acknowledgements

We thank Innovate UK Edge for providing funding to allow this research to be carried out.

## Author contributions

D.J.L.: Devised concept of the study, study design, coordination, first draft and approval of manuscript. R.M.B.: Sample collection, explant analysis, data collection, review and approval of manuscript. T.J.J.: Initiated original study databases, explant analysis, drafting and approval of manuscript. S.P.R.: Review of statistical methodology and data review. Drafting and approval of manuscript. B.J.W.: Study design, first draft, and approval of manuscript. M.E.N.: Statistical analysis and machine learning algorithm development, drafting, review and approval of manuscript. N.S.: Statistical analysis and machine learning algorithm development, drafting, review and approval of manuscript. B.A.L.: Statistical analysis (genetics), drafting and approval of manuscript. M.B.P. Explant analysis and coordination of study at Australian centre, drafting and approval of manuscript. A.J.S.: Guidance on methodology with respect to peptide modelling and metalloprotein binding. Drafting and approval of final manuscript. S.W.: Data collection, maintenance of study database at centre 1, patient review, approval of manuscript. S.N.: Initiated original study database, analysis of samples, drafting and approval of manuscript. R.R.: Data collection, maintenance of study database in United States, patient review, approval of manuscript. R.H.: Data collection, maintenance of study database in United States, patient review, approval of manuscript. R.D.: Statistical analysis (genetics), drafting and approval of manuscript. E.P.S.: Senior surgeon at United States centre, involved in study design, drafting and approval of manuscript. A.V.F.N.: Senior orthopaedic surgeon at centre 1, initiated original study database, patient review, data interpretation, drafting, and approval of manuscript.

## Competing interests

The authors declare the following competing interests: the algorithm described in this study has been developed into software to be used as a commercial medical device (Orthotype). Orthotype has been patented, and is owned by the company PXD Ltd, trading as ExplantLab. David Langton, the lead author, is director of this company. Matthew Nargol is an employee of ExplantLab. All other authors have no competing interests to declare.
