## [Peer Review File · Communications Medicine]

Reviewers' comments:

Reviewer #1 (Remarks to the Author):

To the authors

The manuscript was difficult to review due to missing line numbers, unusual structuring, and misplaced figure captions

In the presented work, the authors investigated the influence of the HLA genotype on the development of arthroprosthetic delayed metal hypersensitivity (DTH). Arthroprosthetic DTH is a topical issue and the identification of potential risk factors for the development of arthroprosthetic DTH is very important to keep patient safety at high level. The analyses by Langton et al. may help to better understand the pathomechanism of arthroprosthetic DTH.

A real strength of the work is its novelty and the stratification of patient cohorts, especially the fact that the effects are controlled for actual metal exposure. This is something that is lacking in many publications dealing with adverse effects associated with arthroprosthetic metal exposure.

Group1: MoM, aggressive ALVAL, low wear

Group2: MoM, asymptomatic at > 10 year follow-up

Group3: MoM, ALVAL, high wear

Group4: MoM, failure w/o ALVAL

The authors found that patients carrying specific HLA-DQ haploid genotypes are prone for developing DTH. Furthermore, the authors validated their findings in a patient cohort (n=606) by developing and training of machine learning models. Such models might be used for patient individual implant choice even though the use of CoCrMo in primary hip arthroplasty is outdated (due to adverse effects related to MoM, the paper shows it). However, the model might be applicable for prognostics in revision hip arthroplasty or knee arthroplasty.

The major claims of the paper are scientifically sound and novel. The work has a distinct translational character and the topic is clinically relevant. It will peak the interest of many professionals including orthopaedic clinicians and researchers. The presented manuscript may deserve publication in Communications Medicine following major revision. Generally, the authors should make more effort to present the results and thoroughly revise the display items (Figures and Data Tables) and add more displayed results in the body of the manuscript.

The following revisions are suggested:

- Formatting and structuring of the manuscript according to the journal requirements
- Abstract: Please specify that the paper focusses on DTH in hip arthroplasty and briefly describe patient groups.
- Introduction: CoCrMo is indeed a material which is used in orthopaedics all over the world on a daily basis but please indicate that the use of CoCrMo in primary hip arthroplasty is outdated and that C-C and C-PE articulations show very good, and compared to hip arthroplasty implants containing CoCrMo, better clinical results.
- Introduction: On page 5 you refer to figure 1. "patients display varying tolerances to metallic

debris". I don't see this information in Figure 1 (If the figure with the explanted resurfacing implants is figure 1)

-Figure 1 (the figure with explanted prostheses and histo image). Please revise this figure and figure caption thoroughly. No need for text within the figure (transfer to Fig. caption). Add a higher magnification H&E image and describe the images. Add scale bars or indicate magnifications.

-Generally, Figs 1-3 should not be part of the introduction section. Fig1 shows results (histo and wear rate). Fig. 2 and 3 could be merged to one Figure.

-The caption of Figure 2 is structured in a very unusual way.

-Please introduce all abbreviations e.g. THR

-A major and very important finding of the work is that Co release per se is a strong causal variable of DTH development. Please perform data analyses in this context and add a respective figure in the result section. I suggest correlating systemic Co levels and the frequency of DTH development.

-Methods: Please add data regarding patient characteristics with a specific focus on implant status.

-Please discuss if there is a basis for using CoCrMo in hip arthroplasty. Are there any known ALVAL associated to hip arthroplasty implants without CoCrMo component? Is the release of Co per se the clinical problem? Is there a need for reintroducing CoCrMo in primary THA?

-Please discuss the role of the peri-implant bone marrow in the context of DTH

I enjoyed reading this manuscript

With kind regards

Reviewer #2 (Remarks to the Author):

The unblinded, presented manuscript describes a targeted genetic (HLA) analysis linking individual patient intrinsic genotype with phenotypical presentation of ALVALs (or lack thereof) around in situ total hip replacements. Genuinely "international" in context, the authors present the results of 606 patients with clinical, intraoperative/pathological, and genetic data from selected cohorts in the United Kingdom, United States and Australia. Some more information about the nature of the host institutions (i.e. quaternary referral centres ?) would allow international readers context. The work appears to be a blend of basic literature review, technique description/validation, and retrospective outcome linkage to prospectively collected genetic data. In keeping with the current state of understanding and evidence base in the available literature, the authors describe focusing potential explanatory gene frequency testing on specific HLA domains with a known association to delayed hypersensitivity responses. There are certainly (many) novel elements to this research of which I found the association between peptide - metal iron binding sites most exciting.

Overall, the work does claim appropriate prospective human research ethics committee approval; is novel in its fundamental nature and premise; includes appropriate (albeit complex) statistical analyses; and the conclusions reached do appear supported by the presented data. For the most part, the paper is well written as presented although there are a number of minor grammatical, presentation, and style errors which I suspect can be easily corrected. Although these – in the current form – do undermine the readability of the paper, they do not alter the underlying solid scientific premise. A large amount of data is presented in the numerous figures and tables at the end of the provided document. I suspect the inclusion of all of these would overcomplicate and burden the paper itself although they would very appropriately be included in most instances as easily accessible appendices. Although I appreciate it is specific to the target journal, I did feel a strong "concluding" statement would add value to the paper itself.

The hope that it may also prove of some value to the authors, I also offer the following individualised feedback:

1. the abbreviation "OECD" should be defined.
2. In the second paragraph of page 3 the authors suggest articulation of the femoral head against a plastic "cup". I would encourage them to consider being specific here as to whether or not this infers a plastic "liner" or and all polyethylene acetabular component, per se. This has different implications for different surgeons around the world and should be made clear. In this vein, I would encourage the authors to scan through the entire document and make sure that the terminology they use is appropriate and consistent in this regard. I would suggest the term "liner" is perhaps a better option.
3. At the end of the second paragraph on page 3 the authors infer that the need for prosthetic component revision infers "failure". I strongly disagree with this and would recommend the authors consider rewording this important statement. In many instances our implants from 15 or 20 years ago require polyethylene exchange due to progressive wear over many years. We would certainly not consider these are failure but rather would champion the good performance of the non-cross-linked polyethylene from this era having survived and serve the patients so well.
4. Paragraph 4 on page 3 which begins "unfortunately, it became ..." is clumsy and should be reworded. The terminology employed is largely nonscientific and appears conversational in nature. Although the sentiment you convey here is a very important one in the context of metal on metal bearings, the wording here has let you down.
5. Page 4 – 'et al' is an abbreviation and should be followed by a full stop. That is, 'et al.' rather than 'et al'. Please scan through the entire document to make sure that this important consideration is applied appropriately throughout.
6. Page 4 - the inclusion of the year of publication in rounded brackets following an in text citation allows the reader to immediately appreciate the recency of the cited work without having to break the flow of reading to refer to the reference list. Please consider including the year of publication in rounded brackets after each in text citation. For example, please consider replacing 'Willert et al' with 'Willert et al. (2005)'. All
7. Page 4 - it is generally considered poor grammatical form in scientific text to begin a sentence with an abbreviation (i.e. 'ALVAL' here). Please scan through the entire document to make sure that this common grammatical rule has been appropriately followed throughout. In instances where you have begun a sentence with an abbreviation please consider either using the expanded form or rewording the sentence such that this important rule is not broken.
8. Please consider softening 'CoCr components leads to ...' to 'CoCr components may lead to ...'.
9. The inclusion of embedded figures within the main body of the manuscript for review is unusual. I would recommend you consider adding the figure and the associated caption on a single page per figure at the end of the document.
10. Generally the descriptors 'figure' and 'table' should be included in a capitalised form throughout the manuscript. That is, 'Figure' and 'Table'. Please scan through the document and make this change is appropriate throughout.
11. All on page 9 who acknowledge a response rate of "around" 60%. More commentary/information must be provided about the 40% of patients who did not respond. Were any statistical analyses performed to determine whether exclusion of this large portion of your eligible cohort may have unjustly biased results.
12. The term "sex" has different connotations and inferred meanings in different parts of the world. Within scientific text, you would be well advised to consider substituting this word with "gender" throughout the document (including figures and tables).
13. Page 23, manufacturer details should be provided for the Thermo Fisher Qubit assay kit.

14. It is generally considered poor scientific form in the main body of a manuscript to include single sentence paragraphs. Page 23 concludes with three of these in the succession. Can these individual sentences either be expanded or merged into nearby paragraphs such that the script rule is followed.

15. General comment - while I do feel that the statistical analyses you have included are appropriate for your detailed analysis, I would implore you to consider your target audience and whether or not these would be optimally appropriate for them. The nature of the work itself would certainly lend itself with potential great interest to orthopaedic surgeons across the planet however I would humbly suggest that the statistical testing described may far exceed the complexity that many such individuals would be comfortable with.

16. Page 11 – 'ICI' has not been introduced to this point in the text and should be done so here. Similarly, at page 28 where the full text (expanded) form of this abbreviation is first shown this should be removed and replaced simply with 'ICI'. This scan through the entire document to ensure that abbreviations are appropriately introduced at first use and then applied consistently thereafter so long as doing so does not involve the abbreviated form beginning a sentence.

17. The manuscript could certainly benefit from a strong summarising/concluding statement at the end of the document.

Reviewer #3 (Remarks to the Author):

This is an interesting study mainly because of the very comprehensive data collection and well explained methodology. There is some question of the a study predicting failure of metal on metal hips as we do not put in MOM hips anymore. But the study can lay the foundation for other HLA studies. Here are my thoughts:

Background –

- There should be some discussion here about the use of metal on metal hips today. Specifically how they are not used pretty much at all anymore because there was an incredibly high incidence of these complications.
- You should also discuss at some point in here, the reason why metal reactions are important today e.g. because of trunionosis, in metal resurfacing, and in dual mobility heads, there is still some metal reaction and therefore this study may be relevant. But saying “unfortunately, it became apparent some patients were beginning to experience problems” is a huge understatement given that some studies show that 75-100% of patients fail MOM hips. It almost makes this study not clinically applicable at all if you do not mention the reasons for metal problems today (e.g. trunionosis and dual mobility metal corrosion) as MOM hips are not put in today. Hip resurfacing does not have the same failure rate and may not be equally applicable either.

Results

- Interesting methodology and statistical analysis
- Really well done analysis and well-written methodology

Discussion

- There needs to be an explanation of the difference in MOM hips and hip resurfacing in more detail. Hip resurfacing is a completely different beast that has a much lower incidence of metal reaction in hip resurfacing.
- Clinical implications: I personally think that this section on TKA is not very applicable. They mention

“there is a lack of consensus as to the clinical significant of metal sensitivity in the field of knee surgery.” What is being left out here is the complete difference in mechanics and wear between TKA and THA. Even if metal sensitivity is important in TKA, which it in itself is debatable - a paper on MOM hips is not relevant in TKAs with metal components but no metal on metal friction or bearing surface. If they want to make this point, they need to make that clear.

General comments to the reviewers

Thank you for taking the time to review our work.

We would like to apologise for the issues with the formatting, which clearly caused issues for the reviewers in the review process.

The article was originally formatted to the "Nature" guidelines – which mandates a very tight word limit, is limited to only four figures/tables, and the methods section is to be required to be placed at the end of the article. We were surprised when Nature expressed interest in review, but eventually the editors recommended (given the subject) it be forwarded on to Comms Medicine as it was felt it would be a better fit for the subject matter.

We were advised that the transfer could be made without any formatting changes needing to be made. We apologise once again for the inconvenience to reviewers we have inadvertently caused. We have tried our best to improve the presentation of the results, with inclusion of tables and extra figures in the main results sections. The Kaplan Meier survival charts we believe are best suited to allow orthopaedic surgeons to interpret the overall results most easily.

In addressing the reviewers' comments the manuscript is lengthy, and the number of figures is high. We were concerned with this, and therefore sought advice from the editorial team at the journal. We were reassured that the word limit is flexible, and that the primary focus should be on addressing reviewers' comments adequately.

There is justified criticism of some of the language and descriptions we have used in the manuscript. We were trying extremely hard to focus on the points we felt most important for a general medical audience. The reason we submitted to a general medical journal was that a broad readership might find the results and their potential application to the development of autoimmune/autoimmune like processes of particular interest, as ALVAL/DTH bears histological and clinical similarities to these conditions.

All reviewers rightly highlight that the use of MoM hip prostheses has largely died out. We felt the research was still of interest because of the above point. But, perhaps more importantly, we felt that the paradox between the widespread cessation of use of CoCr in MoM hips due to tissue reactions, compared to the continued use of CoCr in other arthroplasties was a crucial point to make. The intention is not to scare patients/physicians but to highlight the issues and encourage further studies into the investigation of patients with chronic, unexplained pain following TKRs. We note that only one of the reviewers objects to the inclusion of TKRs in the discussion. While we are very sympathetic with their point, we have not made the link to TKRs without without basis. We have carried out in depth analysis of explanted knee, hips and ankles for several years now. Our latest publication on analysis of explanted contemporary knees reveals significant material loss from CoCr tibial trays – clearly secondary to an abrasive process. Reviewer three also makes the point that the ?friction/mechanisms of wear are very different between MoM hips and knees. In our explant analyses, we do not see such a huge distinction, far from it. Furthermore, in knee revision arthroplasties there is often at least one titanium-CoCr modular junction, sometimes two. The forces transmitted through these junctions have been shown to be equivalent to those in some hip patients. Furthermore, we have no evidence to indicate that CoCr release from knees is less immunogenic than from MoM hips. Blood/serum Co and Cr concentrations in patients with TKRs

with CoCr trays are equivalent to patients with MoM hip resurfacings, and recent in-depth histological studies show that pathological hallmarks of ALVAL/DTH have gone unrecognised in significant percentages of patients who have experienced failure of their TKRs.

In terms of making more detailed comments regarding the viability of MoM hip resurfacings, one could argue that tightening the selection criteria by skimming off the patients most prone to ALVAL might make a significant difference. But we are extremely mindful of the controversies in this area, and passions often take precedent over evidence or lack thereof and we did not want to distract from the results.

Our changes have been highlighted in bold. Please note that the methods section is all highlighted bold – simply because it has been moved from the end of the manuscript to the middle. Any text changes in here are underlined in addition to bold font.

Specific Responses to Reviewer #1

The manuscript was difficult to review due to missing line numbers, unusual structuring, and misplaced figure captions

We apologise for this which was not our intention to disregard Nature Comms Medicine's instructions for authors. Please refer to general comments above.

Generally, the authors should make more effort to present the results and thoroughly revise the display items (Figures and Data Tables) and add more displayed results in the body of the manuscript.

The following revisions are suggested:

-Formatting and structuring of the manuscript according to the journal requirements

Apologies for this, please see general responses above.

-Abstract: Please specify that the paper focusses on DTH in hip arthroplasty and briefly describe patient groups.

This has now been clarified.

-Introduction: CoCrMo is indeed a material which is used in orthopaedics all over the world on a daily basis but please indicate that the use of CoCrMo in primary hip arthroplasty is outdated and that C-C and C-PE articulations show very good, and compared to hip arthroplasty implants containing CoCrMo, better clinical results.

We have tried to address this more satisfactorily now.

-Introduction: On page 5 you refer to figure 1. “patients display varying tolerances to metallic debris”. I don’t see this information in Figure 1 (If the figure with the explanted resurfacing implants is figure 1)

Figure 1 has now been modified and hopefully improved.

-Figure 1 (the figure with explanted prostheses and histo image). Please revise this figure and figure caption thoroughly. No need for text within the figure (transfer to Fig. caption). Add a higher magnification H&E image and describe the images. Add scale bars or indicate magnifications.

Figure 1 has now been modified and hopefully improved.

-Generally, Figs 1-3 should not be part of the introduction section. Fig1 shows results (histo and wear rate). Fig. 2 and 3 could be merged to one Figure.

-The caption of Figure 2 is structured in a very unusual way.

Re the above comments- yes we agree this is unusual. But to some extent this is an unusual study, over a large timeframe which details processes which may be unfamiliar to many readers, and definitely to orthopaedic surgeons. Could we ask for journal editor advice on this? We believe that the images are key to helping the reader understand the concepts and therefore the interpretation of the results.

-Please introduce all abbreviations e.g. THR

We have now addressed this.

-A major and very important finding of the work is that Co release per se is a strong causal variable of DTH development. Please perform data analyses in this context and add a respective figure in the result section. I suggest correlating systemic Co levels and the frequency of DTH development.

The cox proportional hazards models have been brought into the main manuscript and the Kaplan Meier charts should, we hope, make the point of the importance of metal ion levels in relations to the development of DTH.

-Methods: Please add data regarding patient characteristics with a specific focus on implant status.

We have moved data from the supplementary section into the main results which we feel should hopefully address this issue?

-Please discuss if there is a basis for using CoCrMo in hip arthroplasty. Are there any known ALVAL associated to hip arthroplasty implants without CoCrMo component? Is the release of Co per se the clinical problem? Is there a need for reintroducing CoCrMo in primary THA?

We have tried to discuss this to an extent here, but we are conscious of the length of the manuscript and are somewhat limited in doing this full justice.

-Please discuss the role of the peri-implant bone marrow in the context of DTH

Apologies but we feel this would be too much of stretch to go into in the current paper.

Specific Responses to Reviewer #2

1. the abbreviation “OECD” should be defined.

This has now been defined.

2. In the second paragraph of page 3 the authors suggest articulation of the femoral head against a plastic “cup”. I would encourage them to consider being specific here as to whether or not this infers a plastic “liner” or and all polyethylene acetabular component, per se. This has different implications for different surgeons around the world and should be made clear. In this vein, I would encourage the authors to scan through the entire document and make sure that the terminology they use is appropriate and consistent in this regard. I would suggest the term “liner” is perhaps a better option.

We have replaced with “cup” or “liner” as we felt the most important point was that it was the bearing we were referring to – and all PE components or modular acetabular components would essentially be equivalent for the point we were making. We hope this is acceptable.

3. At the end of the second paragraph on page 3 the authors infer that the need for prosthetic component revision infers “failure”. I strongly disagree with this and would recommend the authors consider rewording this important statement. In many instances our implants from 15 or 20 years ago require polyethylene exchange due to progressive wear over many years. We would certainly not consider these are failure but rather would champion the good performance of the non-cross-linked polyethylene from this era having survived and serve the patients so well.

We have modified this to remove the term “failure”.

4. Paragraph 4 on page 3 which begins “unfortunately, it became ...” Is clumsy and should be reworded. The terminology employed is largely nonscientific and appears conversational in nature. Although the sentiment you convey here is a very important one in the context of metal on metal bearings, the wording here has let you down.

We have attempted to correct this.

5. Page 4 – ‘et al’ is an abbreviation and should be followed by a full stop. That is, ‘et al.’ rather than ‘et al’. Please scan through the entire document to make sure that this important consideration is applied appropriately throughout.

This has now been corrected.

6. Page 4 - the inclusion of the year of publication in rounded brackets following an in text citation allows the reader to immediately appreciate the recency of the cited work without having to break the flow of reading to refer to the reference list. Please consider including the year of publication in rounded brackets after each in text citation. For example, please consider replacing ‘Willert et al’ with ‘Willert et al. (2005)’.

While we appreciate this comment, we have tried to keep the referencing in line with the guidance for authors. We would be happy to change should the journal and reviewer wish us to?

7. Page 4 - it is generally considered poor grammatical form in scientific text to begin a sentence with an abbreviation (i.e. ‘ALVAL’ here). Please scan through the entire document to make sure that this common grammatical rule has been appropriately followed throughout. In instances where you have begun a sentence with an abbreviation please consider either using the expanded form or rewording the sentence such that this important rule is not broken.

Thank you we hope we have now corrected this.

8. Please consider softening ‘CoCr components leads to ...’ to ‘CoCr components may lead to ...’.

Modified as suggested.

9. The inclusion of embedded figures within the main body of the manuscript for review is unusual. I would recommend you consider adding the figure and the associated caption on a single page per figure at the end of the document.

Please see general comments at the top, we have removed the figures to the end of the manuscript now.

10. Generally the descriptors ‘figure’ and ‘table’ should be included in a capitalised form throughout the manuscript. That is, ‘Figure’ and ‘Table’. Please scan through the document and make this change is appropriate throughout.

This has been corrected.

11. All on page 9 who acknowledge a response rate of “around” 60%. More commentary/information must be provided about the 40% of patients who did not respond. Were any statistical analyses performed to determine whether exclusion of this large portion of your eligible cohort may have unjustly biased results.

Information added as requested.

12. The term “sex” has different connotations and inferred meanings in different parts of the world. Within scientific text, you would be well advised to consider substituting this word with “gender” throughout the document (including figures and tables).

We understand the sentiment, and we have received conflicting advice in our previous work, having originally used the term “gender”. We settled on “sex” as review of Nature Comms genetic work seems to use “sex” (eg. Campos et al Understanding genetic risk factors for common side effects of antidepressant medications.) We welcome further guidance on this.

13. Page 23, manufacturer details should be provided for the Thermo Fisher Qubit assay kit.

Added as requested.

14. It is generally considered poor scientific form in the main body of a manuscript to include single sentence paragraphs. Page 23 concludes with three of these in the succession. Can these individual sentences either be expanded or merged into nearby paragraphs such that the script rule is followed.

This has been corrected as suggested.

15. General comment - while I do feel that the statistical analyses you have included are appropriate for your detailed analysis, I would implore you to consider your target audience and whether or not these would be optimally appropriate for them. The nature of the work itself would certainly lend itself with potential great interest to orthopaedic surgeons across the planet however I would humbly suggest that the statistical testing described may far exceed the complexity that many such individuals would be comfortable with.

Thank you for the comment, please see our general comments. We struggled with this a great deal, weighing up the extent to which we needed to demonstrate that the algorithm had been tested sufficiently versus the risk of unreadability for clinicians. We have now added in multiple Kaplan Meier charts, which - given their use in national joint registry reports – will hopefully convey the information most effectively.

16. Page 11 – ‘ICI’ has not been introduced to this point in the text and should be done so here. Similarly, at page 28 where the full text (expanded) form of this abbreviation is first shown this should be removed and replaced simply with ‘ICI’. This scan through the entire document to

ensure that abbreviations are appropriately introduced at first use and then applied consistently thereafter so long as doing so does not involve the abbreviated form beginning a sentence.

We have corrected this.

17. The manuscript could certainly benefit from a strong summarising/concluding statement at the end of the document.

Again, we struggled with this and would welcome guidance. We have now added a concluding statement, but we feel whatever we are to write here risks upsetting some part of the orthopaedic community!

Specific Responses to Reviewer #3

Background –

- **There should be some discussion here about the use of metal on metal hips today. Specifically how they are not used pretty much at all anymore because there was an incredibly high incidence of these complications.**

Please refer to the general responses, above.

- **You should also discuss at some point in here, the reason why metal reactions are important today e.g. because of trunionosis, in metal resurfacing, and in dual mobility heads, there is still some metal reaction and therefore this study may be relevant. But saying “unfortunately, it became apparent some patients were beginning to experience problems” is a huge understatement given that some studies show that 75-100% of patients fail MOM hips. It almost makes this study not clinically applicable at all if you do not mention the reasons for metal problems today (e.g. trunionosis and dual mobility metal corrosion) as MOM hips are not put in today. Hip resurfacing does not have the same failure rate and may not be equally applicable either.**

Please refer to general comments above. We hope we have now addressed this.

Discussion

- **There needs to be an explanation of the difference in MOM hips and hip resurfacing in more detail. Hip resurfacing is a completely different beast that has a much lower incidence of metal reaction in hip resurfacing.**

Please refer to general comments above, we hope to have addressed this important point.

• **Clinical implications: I personally think that this section on TKA is not very applicable. They mention “there is a lack of consensus as to the clinical significant of metal sensitivity in the field of knee surgery.” What is being left out here is the complete difference in mechanics and wear between TKA and THA. Even if metal sensitivity is important in TKA, which it in itself is debatable - a paper on MOM hips is not relevant in TKAs with metal components but no metal on metal friction or bearing surface. If they want to make this point, they need to make that clear.**

Thank you for this extremely important comment. Could we refer you to the general comments at the top of our responses?

Reviewers' comments:

Reviewer #1 (Remarks to the Author):

To the authors

I reviewed the revised version of the manuscript THE INFLUENCE OF HLA GENOTYPE ON THE DEVELOPMENT OF METAL HYPERSENSITIVITY FOLLOWING JOINT REPLACEMENT.

The manuscript has improved significantly. Reviewers' comments and suggestions were either implemented through changes in the manuscript or through comprehensive discussion in the point-by-point response. The work is clinically relevant and improves our understanding of DTH in arthroplasty.

One issue remains:

In the first review, I asked whether Co release per se could be a strong causal variable for DTH development. What are the results derived from figures 5 and 7, and what is the point? Is the point that future patients should be genotyped before receiving implants that are prone to Co release and that only patients with a low-risk genotype should receive these implants? Or is it that your analyses suggest that Co exposure itself is the problem and that implants prone to Co release should be avoided. Data in Fig. 5 and Fig. 7 indicate that the probability of implant survival decreases over time (please label y axes) if patients have high systemic Co levels, and that systemic Co levels have a greater impact on implant survival than genotype.

Please communicate a respective outlook in the conclusion section.

On page 21 line 507 to 509 you state that genotyping could be used to further improve the results of resurfacing. Please avoid this statement since resurfacing implants are prone to Co release and Co release has (according to your data) greater impact on implant survival than genotype. Please also keep in mind that Co exposure can lead to various adverse effects.

This discussion is only possible because you have controlled for the actual systemic Co exposure. Your results show the importance of doing this, and in my opinion, this substantially increases the impact of your work.

The manuscript deserves publication after minor revision.

Reviewer #3 (Remarks to the Author):

No further comments

Response to Reviewers' comments on:
**THE INFLUENCE OF HLA GENOTYPE ON THE DEVELOPMENT OF METAL HYPERSENSITIVITY
FOLLOWING JOINT REPLACEMENT.**

Reviewer #1 (Remarks to the Author):

To the authors

I reviewed the revised version of the manuscript THE INFLUENCE OF HLA GENOTYPE ON THE DEVELOPMENT OF METAL HYPERSENSITIVITY FOLLOWING JOINT REPLACEMENT.

The manuscript has improved significantly. Reviewers' comments and suggestions were either implemented through changes in the manuscript or through comprehensive discussion in the point-by-point response. The work is clinically relevant and improves our understanding of DTH in arthroplasty.

One issue remains:

In the first review, I asked whether Co release per se could be a strong causal variable for DTH development. What are the results derived from figures 5 and 7, and what is the point? Is the point that future patients should be genotyped before receiving implants that are prone to Co release and that only patients with a low-risk genotype should receive these implants? Or is it that your analyses suggest that Co exposure itself is the problem and that implants prone to Co release should be avoided. Data in Fig. 5 and Fig. 7 indicate that the probability of implant survival decreases over time (please label y axes) if patients have high systemic Co levels, and that systemic Co levels have a greater impact on implant survival than genotype.

Please communicate a respective outlook in the conclusion section.

On page 21 line 507 to 509 you state that genotyping could be used to further improve the results of resurfacing. Please avoid this statement since resurfacing implants are prone to Co release and Co release has (according to your data) greater impact on implant survival than genotype. Please also keep in mind that Co exposure can lead to various adverse effects.

This discussion is only possible because you have controlled for the actual systemic Co exposure. Your results show the importance of doing this, and in my opinion, this substantially increases the impact of your work.

The manuscript deserves publication after minor revision.

General response

We are thankful to the reviewers for their insightful comments which have helped us improve the manuscript. We are also grateful to the journal for giving us the opportunity to revise the paper.

We submitted this paper to a general science journal (originally Nature), as we believed the interaction between patient gender, genotype and antigen exposure (metal debris) would be of interest (and potentially of relevance in the study of autoimmune/autoimmune like disorders) to a

general audience. We welcomed the manuscript being passed on to Nature Comms Medicine as this would help us reach a general medical audience.

We firmly believe that we have presented the data appropriately to convey the fundamental message of the paper: that the development of metal sensitivity is determined by an interaction between genetics and the prosthetic environment. It is clear that certain genotypes place patients at greater risk of developing DTH at equivalent levels of metal exposure. We are encouraged that the reviewer surmises: *“The work is clinically relevant and improves our understanding of DTH in arthroplasty.”*

In response to the specific comments from the reviewer:

“What are the results derived from figures 5 and 7, and what is the point? Is the point that future patients should be genotyped before receiving implants that are prone to Co release and that only patients with a low-risk genotype should receive these implants? Or is it that your analyses suggest that Co exposure itself is the problem and that implants prone to Co release should be avoided.”

We believe that the results shown in figures 5 and 7 (in combination with figure 6, and the statistical modelling presented in table 2 and supplementary data table 2) show that the risk of DTH is determined by an interaction between genotype, sex and metal levels – the main point of the research and this manuscript. We presented, the results, in part, in the Kaplan Meier charts. We did this because Kaplan Meier charts are the preferred formats for the presentation of results of orthopaedic implants and they are well recognised and understood by professionals in several medical and surgical specialties. The inevitable risk with KM charts, however, is that they may present an over simplistic interpretation of the data. However, we do not see any way around this, and believe that the presentation of the data - taken as a whole - is adequate, allowing the reader to make their own conclusions.

With regard to our commentary on what the results actually mean from a clinical perspective, this is a complex issue and we have discussed this in greater detail below.

The reviewer also stated that:

*“Data in Fig. 5 and Fig. 7 indicate that the probability of implant survival decreases over time if patients have high systemic Co levels, and **that systemic Co levels have a greater impact on implant survival than genotype.**”*

We disagree with the reviewer’s statement, feeling that it is an over simplification. The relative effects of the interacting variables are shown in figures 5, 6 and 7 and in the modelling in table 2 and supplementary data table 2.

One simply cannot state “Co levels have a greater impact” without putting normally observed variation in metal ion concentrations into context. We have studied the relationship between prosthetic wear and rates and metal ion levels for over 15 years.[1-6] With regard to hip resurfacing implants, a low wearing device (<1mm³ per year in volumetric terms) equates to a Co concentration of, in general, less than 1.5 micrograms per litre.[7, 8] If a resurfacing device malfunctions to a modest extent, and doubles in wear, Co levels increase to approximately 3.5 micrograms per litre; a three fold elevation in wear is associated with Co level of around 5 micrograms per litre. It is within these levels that well designed, well positioned devices will almost always perform.[5, 9] It is to be

noted that this study included the ASR hips, which is well recognised to wear at a much higher rate, leading to enormous blood metal concentrations – above 200 micrograms per litre.[8, 10, 11] Even with the ASR included, around 70% of all the patients in this study had blood Co concentrations less than 3 micrograms per litre. The median and range of blood metal ion concentrations are presented in the manuscript. We also note that we have dedicated a whole section in the discussion to the subject of variation in metal ion levels in patients with hip resurfacings and knee replacements. Put most simply, for a given metal ion concentration, higher risk genotypes confer approximately a three-fold increase risk of DTH.

The reviewer objects to our comment that genotyping could be used to improve the results of resurfacing: *“On page 21 line 507 to 509 you state that genotyping could be used to further improve the results of resurfacing. Please avoid this statement since resurfacing implants are prone to Co release and Co release has (according to your data) greater impact on implant survival than genotype.”*

Our comment in the manuscript under review was: *“It now seems possible that genotyping could be used to further improve the results of resurfacing.”* We believe this statement is accurate and appropriate given the evidence that we have presented in the paper.

Again, we disagree with the reviewer’s statement that *“Co release has greater impact on implant survival than genotype”* for the reasons given above. If a hip resurfacing wears at an expected rate, with the corresponding blood level of less than 2 micrograms per litre, the risk of ARMD in males is 1.8% at twelve years and 0% for females if the patients do not have DQA1*02:01-DQB1*02:02. In patients with this gene present, the risks increase to 6.9 and 15.4% respectively. One might argue that the confidence intervals just overlap, but the difference is clearly significant when all variables are incorporated, and ion levels are used as the continuous variables that they are in reality. The use of KM charts is also not ideal here as risk genotypes are not “present” or “absent” – they are a hierarchy. The Cox proportional hazards modelling take these factors into account – detailed in the tables – but this is relatively complex, and not readily understood by most readers, hence why we added multiple KM charts.

The reviewer states that *“resurfacing implants are prone to Co release”*. Yet as we discuss in the paper almost every knee replacement in the world involves implantation of a CoCr component.[12, 13] ANY CoCr implant is prone to Co release (or chromium, or whatever material the implant is composed of) once it enters the body – it is simply a matter of magnitude due to the inevitable wear and corrosive effects brought about by movement and exposure to body fluids.[14-17]

So if we are to recommend the avoidance of hip resurfacings due to the fact that they are prone to releasing Co, we would be recommending the cessation of almost all knee replacements commonly used around the world. Clearly this would neither be sensible, but neither would it be based on the data we have. Our aim was to place everything in context, it is for the journal to decide whether that was successful.

We suspect it is likely that the reviewer is meaning that *“hip resurfacings are prone to releasing more Co”* and therefore should be avoided. We would agree this is true, but could be seen as misleading. We have put the amount of Co release into context in the last section of the discussion. Median hip resurfacing metal ion levels are comparable - if not lower - than those seen in patients with TKRs with CoCr components.[18-20] Gross increases in wear of hip resurfacing prostheses are

generally secondary to surgical malpositioning[21, 22], and with robotic surgery increasing in prevalence it is thought that gross errors in component orientation will become less frequent.

We understand the reviewer's comments and believe they are valid. However, we do not want to go beyond the purpose of the manuscript. All joint replacements and joint replacement materials carry with them advantages and disadvantages, but we feel that a thorough discussion warrants its own paper, and would be more appropriate in an orthopaedic or specialist arthroplasty journal.

We reiterate that the purpose of the manuscript was to demonstrate the interacting variables leading to DTH following joint replacement. We do not feel it is justified or reasonable to go on to recommend a blanket cessation of use of CoCr implants. Nor do we feel it reasonable to suggest that all patients should undergo genotyping, we are simply trying to make clinicians aware of how these variables interact and provide more evidence as to the risks and mechanism leading to DTH, which is a very poorly understood phenomenon.[23-27]

1. Langton, D., et al., *The effect of component size and orientation on the concentrations of metal ions after resurfacing arthroplasty of the hip*. The Journal of bone and joint surgery. British volume, 2008. **90**(9): p. 1143-1151.
2. Langton, D.J., et al., *Blood metal ion concentrations after hip resurfacing arthroplasty: a comparative study of articular surface replacement and Birmingham Hip Resurfacing arthroplasties*. J Bone Joint Surg Br, 2009. **91**(10): p. 1287-95.
3. Jameson, S., et al. *REDUCING EXPOSURE TO METAL IONS FOLLOWING HIP RESURFACING: THE IMPORTANCE OF ACETABULAR ORIENTATION*. in *Orthopaedic Proceedings*. 2011. The British Editorial Society of Bone & Joint Surgery.
4. Langton, D.J., et al., *Reducing metal ion release following hip resurfacing arthroplasty*. Orthop Clin North Am, 2011. **42**(2): p. 169-80, viii.
5. Sidaginamale, R., et al., *Blood metal ion testing is an effective screening tool to identify poorly performing metal-on-metal bearing surfaces*. Bone & joint research, 2013. **2**(5): p. 84-95.
6. Sidaginamale, R.P., et al., *The clinical implications of metal debris release from the taper junctions and bearing surfaces of metal-on-metal hip arthroplasty: joint fluid and blood metal ion concentrations*. Bone Joint J, 2016. **98-b**(7): p. 925-33.
7. Joyce, T., D. Langton, and A. Nargol. *The wear of ex vivo metal-on-metal resurfacing hip prostheses*. in *Orthopaedic Proceedings*. 2011. The British Editorial Society of Bone & Joint Surgery.
8. Langton, D., et al., *Accelerating failure rate of the ASR total hip replacement*. The Journal of bone and joint surgery. British volume, 2011. **93**(8): p. 1011-1016.
9. Langton, D., *Are Metal Ion Levels a Trigger for Surgical Intervention?*, in *Metal-on-Metal Bearings*. 2014, Springer. p. 63-82.
10. Langton, D., et al., *Early failure of metal-on-metal bearings in hip resurfacing and large-diameter total hip replacement: a consequence of excess wear*. The Journal of bone and joint surgery. British volume, 2010. **92**(1): p. 38-46.
11. Cohen, D., *Out of joint: the story of the ASR*. BMJ 2011. **342**.
12. listed, N.a., *Australian Orthopaedic Association National Joint Replacement Registry. Annual Report*. Adelaide: AOA. 2016.
13. Partridge, T.C.J., et al., *Conventional Versus Highly Cross-Linked Polyethylene in Primary Total Knee Replacement: A Comparison of Revision Rates Using Data from the National Joint Registry for England, Wales, and Northern Ireland*. J Bone Joint Surg Am, 2020. **102**(2): p. 119-127.

14. Huot Carlson, J.C., et al., *Femoral stem fracture and in vivo corrosion of retrieved modular femoral hips*. J Arthroplasty, 2012. **27**(7): p. 1389-1396.e1.
15. Jani, S.C., et al., *Fretting Corrosion Mechanisms at Modular Implant Interfaces*, D.E. Marlowe, J.E. Parr, and M.B. Mayor, Editors. 1997, ASTM International: West Conshohocken, PA. p. 211-225.
16. Urban, R.M., et al., *Dissemination of wear particles to the liver, spleen, and abdominal lymph nodes of patients with hip or knee replacement*. J Bone Joint Surg Am, 2000. **82**(4): p. 457-76.
17. Kurmis, A.P., et al., *Pseudotumors and High-Grade Aseptic Lymphocyte-Dominated Vasculitis-Associated Lesions Around Total Knee Replacements Identified at Aseptic Revision Surgery: Findings of a Large-Scale Histologic Review*. J Arthroplasty, 2019. **34**(10): p. 2434-2438.
18. Daniel, J., et al., *Results of Birmingham hip resurfacing at 12 to 15 years: a single-surgeon series*. Bone Joint J, 2014. **96-b**(10): p. 1298-306.
19. Luetzner, J., et al., *Serum metal ion exposure after total knee arthroplasty*. Clin Orthop Relat Res, 2007. **461**: p. 136-42.
20. Savarino, L., et al., *The potential role of metal ion release as a marker of loosening in patients with total knee replacement: a cohort study*. J Bone Joint Surg Br, 2010. **92**(5): p. 634-8.
21. Langton, D.J., et al., *The effect of component size and orientation on the concentrations of metal ions after resurfacing arthroplasty of the hip*. J Bone Joint Surg Br, 2008. **90**(9): p. 1143-51.
22. De Haan, R., et al., *Revision of metal-on-metal resurfacing arthroplasty of the hip: the influence of malpositioning of the components*. J Bone Joint Surg Br, 2008. **90**(9): p. 1158-63.
23. Innocenti, M., et al., *Metal hypersensitivity after knee arthroplasty: fact or fiction?* Acta bio-medica : Atenei Parmensis, 2017. **88**(2S): p. 78-83.
24. Granchi, D., et al., *Metal hypersensitivity testing in patients undergoing joint replacement: a systematic review*. J Bone Joint Surg Br, 2012. **94**(8): p. 1126-34.
25. Bravo, D., et al., *No Increased Risk of Knee Arthroplasty Failure in Patients With Positive Skin Patch Testing for Metal Hypersensitivity: A Matched Cohort Study*. J Arthroplasty, 2016. **31**(8): p. 1717-21.
26. Malahias, M.A., et al., *Allergy Testing Has No Correlation with Intraoperative Histopathology from Revision Total Knee Arthroplasty for Implant-Related Metal Allergy*. J Knee Surg, 2021.
27. Haddad, S.F., et al., *Exploring the Incidence, Implications, and Relevance of Metal Allergy to Orthopaedic Surgeons*. J Am Acad Orthop Surg Glob Res Rev, 2019. **3**(4): p. e023.

REVIEWERS' COMMENTS:

Reviewer #1 (Remarks to the Author):

Dear authors

Thank you for the accessible and detailed response. I recommend the publication of the manuscript in Nature Communications Medicine.

Kind regards